# Comparative Study of Selected Excipients’ Influence on Carvedilol Release from Hypromellose Matrix Tablets

**DOI:** 10.3390/pharmaceutics15051525

**Published:** 2023-05-18

**Authors:** Tadej Ojsteršek, Grega Hudovornik, Franc Vrečer

**Affiliations:** 1KRKA, d. d., Novo mesto, Šmarješka cesta 6, 8501 Novo Mesto, Slovenia; 2Faculty of Pharmacy, University of Ljubljana, 1000 Ljubljana, Slovenia

**Keywords:** hypromellose, HPMC, controlled release, modified release, extended release, prolonged release, direct compression, excipient, filler, drug release modifier

## Abstract

Solid dosage forms based on hypromellose (HPMC) with prolonged/extended drug release are very important from the research and industrial viewpoint. In the present research, the influence of selected excipients on carvedilol release performance from HPMC-based matrix tablets was studied. A comprehensive group of selected excipients was used within the same experimental setup, including different grades of excipients. Compression mixtures were directly compressed using constant compression speed and main compression force. LOESS modelling was used for a detailed comparison of carvedilol release profiles via estimating burst release, lag time, and times at which a certain % of carvedilol was released from the tablets. The overall similarity between obtained carvedilol release profiles was estimated using the bootstrapped similarity factor (f_2_). In the group of water-soluble carvedilol release modifying excipients, which produced relatively fast carvedilol release profiles, POLYOXᵀᴹ WSR N-80 and Polyglykol^®^ 8000 P demonstrated the best carvedilol release control, and in the group of water-insoluble carvedilol release modifying excipients, which produced relatively slow carvedilol release profiles, AVICEL^®^ PH-102 and AVICEL^®^ PH-200 performed best.

## 1. Introduction

Hypromellose (HPMC) based hydrophilic matrix tablets are a therapeutically important and widely used controlled-release (CR) dosage form for achieving prolonged/extended drug release. They are designed to hydrate on swallowing, swell and form a ‘gel’ layer of the hydrated HPMC at the tablet surface. The formation of this ‘gel’ layer is responsible for controlling the drug release rate during the passage of the matrix tablet through the gastrointestinal tract (GIT). HPMC is a popular matrix-forming polymer due to its physicochemical characteristics, safety, global compliance, and availability. The 2208 substitution type of HPMC is the most used and has been reported to produce slower drug release profiles than other HPMC types of similar molecular weight. HPMC is available in several viscosity grades, which enables formulators to design matrices with different drug-release mechanisms, i.e., predominantly diffusion, erosion, or mixed diffusion/erosion mechanisms. Direct compression is a simple, low-cost, and suitable method of HPMC matrix tablet manufacture, making it a preferred choice [1,2,3,4,5,6,7].

Several known factors influence the performance of drug release from HPMC matrix tablets. The most important formulation factors are the HPMC substitution type, viscosity grade, and content in tablets, as well as the choice of other excipients (other polymers, fillers, drug release modifiers) included in the tablets. HPMC batch-to-batch consistency is also important because variations in the molecular weight, degree of substitution, and variations in HPMC particle size can influence the drug release performance. Physicochemical characteristics of the drug play an important role in drug release performance, especially drug solubility. On the manufacturing side, the main compression force is recognized as an important factor as well as the size and shape of the tablets [3].

The focus of our research is investigating the influence of selected excipients, which can be used in a high content as filers within the tablets, on the performance of drug release from HPMC matrix tablets. From the literature review, it is apparent that fillers were the most studied excipients in HPMC matrix dosage forms, where, generally, soluble fillers produced faster drug release profiles than insoluble ones. There are some exceptions to this rule found in the literature, where HPMC matrix failure occurred with insoluble fillers, leading to fast drug release. The majority of research was performed using soluble fillers lactose and mannitol and insoluble fillers dibasic calcium phosphate (DCP), microcrystalline cellulose (MCC), and pregelatinised starch [3,6,8,9,10,11,12,13,14,15,16,17,18,19,20,21,22,23,24,25,26,27,28,29].

The main deficiency in the literature regarding the published studies of fillers/excipients’ influence on HPMC matrix performance in general is that a comprehensive selection of excipients was never tested within the same experimental system. There are differences among individual studies in the HPMC substitution type, viscosity grade and content used, differences in the drug used, differences in the tablet shape and size, and differences in the manufacturing procedure and process parameters used, such as the main compression force. The amount of the drug used in the research is not always within the therapeutic range, which makes sense for a once-daily treatment using the chosen drug. Usually, only one representative grade of a filler or excipient is used for each chemical entity of the filler/excipient, and several filler/excipient grades are not researched within the same experimental system. The focus of research found in the literature is mainly on average drug release, whereas the influence of excipient choice on tablet-to-tablet variation is not well covered. In addition, there is no emphasis put on the occurrence of burst release (the uncontrolled and rapid dissolution and release of the drug from the tablets’ surface at the beginning of the drug release profile) or lag time (the delay of the onset of drug release) using different excipients in HPMC dosage forms. Both can have implications for the quality of HPMC tablets as both can lead to unwanted drug release testing results and, in severe cases, failure to meet the drug release specification criteria. Failure to achieve a targeted in vitro drug release profile can result in failure to achieve a targeted in vivo plasma drug release profile, which can have important implications for the safety and efficacy of the dosage form.

The bootstrapped similarity factor (f_2_) is used for a robust and reliable overall determination of significant differences among experimentally obtained drug release profiles. However, f_2_ is a single number estimate and does not provide detailed information regarding where in the drug release profiles differences among different formulations occur, nor does it provide any information regarding the occurrence of burst release or lag time. To compare the obtained drug release profiles in detail, it is useful to calculate the times at which a certain % of the drug is released from the tablets. This information is useful in targeting a desired drug release profile and, at the same time, provides an estimate of burst release or lag time. For this purpose, mathematical modelling of drug release profiles is usually applied using established mathematical drug release models such as Higuchi, Korsmeyer–Peppas, etc. Local (Weighted) Regression (LOESS or LOWESS), a general modelling and signal smoothing method, has never been used as an alternative drug release profile modelling and analysis method, although it can model a broad range of different nonlinear shapes.

In the presented research, a comprehensive selection of excipients was studied regarding their influence on drug release performance from HPMC matrix tablets. Several grades of excipients were tested. The usage of polyethene glycol (PEG), a lower viscosity grade of polyethene oxide (PEO)—lower than that used in CR formulations or combinations with HPMC [30,31]—and povidone in high content in tablets were also tested. Povidone has been used as a drug release modifier in HPMC tablets before [32], but not in such high amounts. No studies of PEGs and lower viscosity grades of PEO as drug release modifiers in HPMC matrix tablets were found in the literature.

Formulation parameters, such as the amount of HPMC, drug, selected excipients/drug release modifiers, and other formulation ingredients, were kept constant (ratios between the mentioned formulation factors were consequently also kept constant), along with the target tablet weight, target main compression force, and the manufacturing equipment used in the experiments. The same type, viscosity grade, and batch of HPMC (HPMC 2208, Methocel K15M) were used for all experiments. Carvedilol (free base) was used as a model drug substance in a 64.8 mg dose per tablet. This is equivalent to 80 mg of carvedilol phosphate per tablet used in the highest strength of the marketed product Coreg CR^®^ extended-release capsules 80 mg, which is designed for once-daily dosing of carvedilol and is therapeutically equivalent to twice-daily dosing of an immediate-release (IR) carvedilol formulation containing 25 mg of carvedilol (free base) per dose [33,34]. Since the marketed formulation is not an HPMC-based matrix system and comes in capsule dosage form, a decision was made to experiment with incorporating carvedilol into directly compressed HPMC-based matrix tablets. Direct compression was chosen as a method of tablet manufacture for two main reasons. Firstly, due to its simplicity and low tablet production cost. Secondly, to be able to fully investigate the influence of chosen excipients on the drug release profiles.

The main research goal is to investigate the influence of a comprehensive selection of excipients on drug release from an HPMC matrix tablet within the same experimental setup and compare the obtained drug release profiles in a detailed way.

## 2. Materials and Methods

### 2.1. Materials

Carvedilol (free base) was supplied by Krka, d.d., Novo Mesto, Slovenia. Refer to Table 1 and Appendix A for function and particle size information, respectively.

Hypromellose (HPMC) of the 2208 substitution type (HPMC 2208) and nominal viscosity of 15,000 mPa·s (cP), METHOCELᵀᴹ K15M Premium, was obtained from DOW, Specialty Electronic Materials Switzerland GmbH, Luzern, Switzerland. The same batch of HPMC was used in all of the experiments. The certificate of analysis (CoA) stated values of the main functionality-related characteristics (FRCs) for the HPMC batch used, which are summarised in Table 1. Particle size information is available in Appendix A.

Silica, Colloidal Anhydrous (Ph. Eur), i.e., Colloidal Silicon Dioxide (USP), AEROSIL^®^ 200 Pharma, was obtained from Evonik Operations GmbH, Germany. Refer to Table 1 for function.

Magnesium stearate, Magnesium stearate EUR PHAR Vegetable, was obtained from FACI S.p.A., Carasco, Italy. Refer to Table 1 and Appendix A for function and particle size information, respectively.

In our experiments, different selected excipients as water-soluble and water-insoluble carvedilol release modifiers were used. The information on the used selected excipients is compiled in Table 2.

Tablet composition and summary of used selected excipients as carvedilol release modifiers are presented in Table 1 and Table 2, respectively. All of the used materials complied with their Ph. Eur. specifications.

### 2.2. Methods

#### 2.2.1. Compression Mixtures Preparation

The compression mixtures were prepared on a laboratory scale in 1 kg batch sizes using a 6 L biconical mixer at 32 rpm mixing speed. The volume fill % was between 1/3 and 2/3 of the available blender volume, dependent on the bulk volume of prepared compression mixtures. Homogenisation of the ingredients before the addition of lubricant (magnesium stearate) was performed for 8 min, and additional homogenisation after the addition of lubricant was performed for 2 min. The homogenization conditions were chosen based on preliminary experiments.

#### 2.2.2. Production of Tablets

Tablets were prepared via direct compression of compression mixtures using the Killian Pressima rotary tablet press and round, slightly concave (R = 30 mm) punches with a 12 mm diameter and bevelled edges. The same compression speed of 7200 tbl/h, the same fill-o-matic speed of 10 rpm, and the same main compression force of 20 kN (with no pre-compression) were used in all of the performed experiments. The main compression force of 20 kN was chosen based on preliminary experiments to achieve satisfactory friability results of tablets. The targeted average weight of tablets was 648 mg. Individual tablet weight variation was low, with maximal observed relative standard deviation % (RSD %) of 1.5% and an average observed RSD % of just 0.6%.

#### 2.2.3. Carvedilol Release Profiles

The carvedilol release profiles were obtained using a method previously described by Košir et al. [35], which proved suitable for carvedilol (free base) release testing where carvedilol is incorporated into HPMC-based matrix tablets. Therefore, a dissolution apparatus type 2 with paddles, flow-through cuvettes and an autosampler, 900 mL of acetate buffer solution (pH = 4.5) per vessel, dissolution media temperature of 37 °C ± 0.5 °C and sinkers to keep tablets at the bottom of the vessel were used. The amount of carvedilol released was spectrophotometrically determined at 285 nm from the measured absorbance and calculated using a calibration curve prepared in advance (carvedilol concentration range 0.00359–0.08985 mg/mL, achieved linear signal response with calibration curve RSQ of 0.99997). Four tablets per experiment were tested. The preselected time points for sampling were the same for all experiments and were as follows: start—0.5 h every 10 min, at 45 min, 1 h–6 h every 30 min, 6 h–24 h every 60 min. Carvedilol release at earlier time points (less than t = 10 min) was not analysed because sampling the first data point at t = 10 min ensured a good signal-to-noise ratio in the utilised dissolution measurement system for all formulations. Sampling too early after dropping the tablets into the vessels could have resulted in carvedilol release results, where a significant portion of the carvedilol release variation would have originated from a poor signal-to-noise ratio and not from the actual tablet-to-tablet carvedilol release variability. The carvedilol release results are presented in Appendix A and Figures 1, 3–7 and Appendix A.

#### 2.2.4. LOESS Modelling and Analysis of Dissolution Data in Excel

LOESS (or LOWESS) is a modern modelling method that is primarily used for smoothing data or signals. It uses local regression, also known as moving regression. For each X value where a Y value is to be calculated, the LOESS technique performs a regression on points in a moving range around the X value, where the values in the moving range are weighted according to their distance from this X value [36,37]. Microsoft Excel VBA code for the LOESS function, which was used in our research, was obtained from the Peltier Tech website [38]. LOESS combines much of the simplicity of linear least squares regression with the flexibility of nonlinear regression. It does this by fitting simple models to localized subsets of the data to build up a function that describes the deterministic part of the variation in the data fairly well [39,40]. In the LOESS application of modelling dissolution data, the X values are time points and Y values were experimentally determined % of carvedilol released at these time points. Using LOESS, re-estimations of the % of carvedilol released at each experimental time point were possible using dissolution data from a subset of experimentally obtained data points around the data point of interest. In addition, the estimation of the % of carvedilol released at any selected time point between the experimentally obtained data points and before the first experimentally measured data point could be performed. A relatively small subset size of five experimentally obtained data points was used per single estimation to maintain a relatively high resolution of the LOESS models. This turned out to be necessary as some carvedilol release profiles were relatively fast for a prolonged-release matrix system, and consequently, changes in carvedilol release were relatively severe from one experimental time point to another. Using LOESS, the % of carvedilol released from t = 0 to t = 1440 min (24 h) with a resolution of 1 min was calculated to produce LOESS carvedilol release profile data. The LOESS carvedilol release data estimates from t = 0 up to and not including t = 10 min are extrapolations from experimental data. From this part of the LOESS carvedilol release models, the estimated % of carvedilol released at t = 0, namely F_0,LOESS_, was used as a burst release indicator/estimator to identify the presence and estimate the extent of burst release (in the case F_0,LOESS_ was significantly >0%, i.e., subtracting one standard deviation from mean F_0,LOESS_ was still above 0% of carvedilol release) and as a lag time indicator to estimate the presence of lag time (in the case estimated F_0,LOESS_ was significantly <0%, i.e., adding one standard deviation to mean F_0,LOESS_ was still below 0% of carvedilol release). Furthermore, the use of LOESS VBA code and Microsoft Excel’s Solver was combined to iteratively estimate times at which 0% (T_0%,LOESS_ i.e., T_lag,LOESS_), 5% (T_5%,LOESS_), 10% (T_10%,LOESS_)… and 90% (T_90%,LOESS_) of carvedilol was released from the tested tablets. The LOESS-estimated T_0%,LOESS_ i.e., T_lag,LOESS_ was used to estimate carvedilol release lag time and was iteratively calculated only if F_0,LOESS_ was significantly <0%. All of these LOESS calculations were performed for each of the four tested tablets per experiment, so the necessary statistics to present results, i.e., average ± standard deviation (SD), could be calculated. The relevant results using LOESS are presented in Appendix A (model fit data), Appendix A (T_X%,LOESS data_), and Figure 2 and Appendix A.

#### 2.2.5. Carvedilol Release Profile Comparison Using Similarity Factor (f_2_)

For each combination of two carvedilol release profiles, the observed similarity factor (f_2_) was calculated and the lower bound of the bootstrapped 90% confidence interval for the similarity factor (i.e., 5th percentile of bootstrapped f_2_ values) was estimated; *n* = 5000 was used in bootstrapping. The results are presented in Appendix A. The f_2_ calculations were performed using DDSolver [41]. Two carvedilol release profiles were considered ‘similar’ if both the observed f_2_ and the 5th percentile of bootstrapped f_2_ values were ≥50.0 [41,42]. In case only the observed f_2_ was ≥50.0 and the 5th percentile of bootstrapped f_2_ values was not, the two carvedilol release profiles were considered as ‘partially similar’. In the latter case, the mean carvedilol release profiles are similar enough to result in observed f_2_ ≥50.0; however, a significant portion (>5%) of bootstrapped profiles fail to meet the f_2_ similarity criteria. In the ‘Results and Discussion’ chapter, the similarity of two carvedilol release profiles is frequently commented as ‘similar’ or ‘partially similar’, where herein established criteria for ‘similarity’ or ‘partial similarity’ of every two compared carvedilol release profiles are strictly adhered to.

#### 2.2.6. Determination of Particle Size Distribution

The particle size distribution of the materials used in experiments was determined with the laser diffraction method using a Malvern Mastersizer 2000 (Malvern, UK). A wet dispersion method was used for carvedilol and a dry dispersion method for all other excipients. The results of particle size distribution were given based on the volume-weight diameter (D_10_, D_50_, D_90_, mean-D_(4,3)_). Colloidal silicon dioxide was not measured since this material was not suited for particle size analysis using the mentioned method. The particle size distribution measurement results are presented in Appendix A.

## 3. Results and Discussion

### 3.1. Carvedilol Release Results, LOESS Modelling, and Carvedilol Release Profile Comparison

#### 3.1.1. Overview

In the present research, a detailed comparison of drug release profiles obtained in presented experiments, not only an overall one using similarity factor (f_2_), was performed. Calculation of times at which a certain % of the drug was released was utilized, as well as obtaining estimates of F_0_, a burst release estimator, and T_lag_, a lag time estimator. At first, the employment of classical and established mathematical models was tried for this type of analysis using DDSolver, but several problems occurred. Firstly, a suitable mathematical model, which produced a reasonably good fit for all the tested tablets per formulation, could not always be found. This made the estimated drug release data between experimentally determined data points and model-dependent re-estimations of experimentally determined data points unreliable. Consequently, the model-dependent estimations of times at which a certain % of the drug was released were also not accurate enough. Secondly, not every mathematical model available in DDSolver had an option of estimating F_0_ and/or T_lag_, which was intended to be employed in the comparison of drug release performance for tested formulations. To solve these issues, an experimental new application of LOESS was tried. LOESS was utilized for modelling of drug release profiles, estimation of F_0,LOESS_ and T_lag,LOESS_ (referred to simply as F_0_ and T_lag_ in the further text), and in the calculation of times at which a certain % of the drug was released, i.e., T_X%,LOESS_ (refer to Section 2.2.4. for explanation). This enabled the comparison of the influence of selected excipients on drug release performance in a more comprehensive, i.e., detailed way, which has not been described in the literature yet.

The results of mean carvedilol release from different formulations containing different selected carvedilol release modifying excipients are summarized in Figure 1. Experimentally obtained data on carvedilol release are available in Appendix A. Generally, all water-soluble excipients resulted in faster carvedilol release profiles than water-insoluble ones. More detailed figures demonstrating mean carvedilol release together with standard deviation (SD) for different groups of selected excipients used as carvedilol release modifiers, namely PEG/PEO group, PVP group, mannitol group, lactose group, sucrose/maltodextrin group, DCP/pregelatinized starch group, and MCC/EC group, is presented further on in the Section 3.

The observed f_2_ and the 5th percentile of bootstrapped f_2_ values were used for the identification of overall significant differences in carvedilol release profiles (see Appendix A). The observed f_2_ is only able to estimate the similarity or dissimilarity of two drug release profiles using mean drug release at each time point, while bootstrapped f_2_ also considers tablet-to-tablet variation in the drug release. Several examples are pointed out further in the discussion where observed f_2_ indicates similarity between two carvedilol release profiles while bootstrapped f_2_ fails to confirm similarity due to one or both of the carvedilol release profiles exhibiting significant tablet-to-tablet carvedilol release variation. Generally, carvedilol release profiles of formulations using selected soluble excipients mostly differ significantly among groups of selected excipients (PEG group, PVP group, lactose group, mannitol group, sucrose, maltodextrin) as well as among different grades of selected excipients within the same group. In contrast, differences in carvedilol release from formulations using selected insoluble excipients are less significant. A more detailed comment on f_2_ results is provided in the later text, where individual groups of selected excipients are discussed.

LOESS models of carvedilol release profiles produced a good fit with experimental data for all individually tested tablets (see Appendix A). This indicates calculated T_X%,LOESS_ results can be confidently utilized for a detailed comparison of obtained carvedilol release profiles (see Appendix A) as well as F_0_ and T_lag_ in comparing the occurrence and extent of burst release and lag time (see Figure 2) among different formulations. The LOESS modelling results are described in more detail in the discussion below.

#### 3.1.2. Formulations Using Soluble Selected Excipients

##### PEG/PEO and PVP

In the PEG/PEO group of formulations, formulations with both PEGs, Polyglykol^®^ 4000 P and Polyglykol^®^ 8000 P, demonstrated faster carvedilol release than the PEO formulation using POLYOXᵀᴹ WSR N-80 (see Figure 3a). According to the f_2_ metric, carvedilol release profiles of both PEG formulations are similar to each other and significantly different from the PEO formulation (see Appendix A). However, the Polyglykol^®^ 4000 formulation’s inter-tablet release variability is larger in comparison to the Polyglykol^®^ 8000 P one (see Figure 3a, Appendix A). The carvedilol release profile of the POLYOXᵀᴹ WSR N-80 formulation was slower than that of both PEG formulations, and variation of carvedilol release among individual tablets was lower than that of the Polyglykol^®^ 4000 P formulation but somewhat higher than that of the Polyglykol^®^ 8000 P one (see Figure 3a, Appendix A). The latter difference is not practically important as the observed variability in carvedilol release from the POLYOXᵀᴹ WSR N-80 formulation is still very low. LOESS analysis of F_0_ and T_lag_ shows that the Polyglykol^®^ 4000 P formulation demonstrates a small F_0_, which indicates the occurrence of a small burst effect, whereas Polyglykol^®^ 8000 P and POLYOXᵀᴹ WSR N-80 formulations demonstrate T_lag_ with the POLYOXᵀᴹ WSR N-80 formulation showing larger T_lag_ than the Polyglykol^®^ 8000 P one (see Figure 2). All these observations are generally in line with differences among these three excipients in molecular weight (POLYOXᵀᴹ WSR N-80 > Polyglykol^®^ 8000 P > Polyglykol^®^ 4000 P), the viscosity of solution (POLYOXᵀᴹ WSR N-80 > Polyglykol^®^ 8000 P > Polyglykol^®^ 4000 P), and the number of ether groups (-O- groups) per single molecule of PEG/PEO (POLYOXᵀᴹ WSR N-80 > Polyglykol^®^ 8000 P > Polyglykol^®^ 4000 P), which are hydrogen bond (H-bond) acceptors and can interact with hydroxyl groups (-OH groups) of HPMC (hydrogens in -OH groups of HPMC acts as H-bond donors). A larger molecular weight of PEG/PEO results in a higher viscosity of solution [43,44] and a higher number of -O- groups per single PEG/PEO molecule potentially available for H-bond interactions with the –OH groups of HPMC, which contributes to slower carvedilol release in comparison to both PEG formulations. Some differences in carvedilol release profiles among these three formulations may also arise from differences in the particle size of selected excipients (see Appendix A) and the mechanical properties of produced tablets.

PVP formulations containing KOLLIDON^®^ 25 and KOLLIDON^®^ 90 F are not similar according to the f_2_ metric (see Appendix A). They both differ in carvedilol release rate, carvedilol release variability, and occurrence of F_0_ and T_lag_. The first two observations are expected because of the significantly different viscosities of both PVP grades’ solutions [45]. Tablets with KOLLIDON^®^ 25 exhibit faster carvedilol release, higher variability of carvedilol release among individual tablets, and non-conclusive F_0_ or T_lag_ in comparison to tablets with KOLLIDON^®^ 90 F, which exhibit T_lag_ (see Figure 2 and Figure 3b, Appendix A). Similar to the PEG/PEO group, the differences in carvedilol release profiles between both PVP formulations can be explained by the differences between used PVP grades in molecular weights (KOLLIDON^®^ 90 F > KOLLIDON^®^ 25), the viscosity of solution (KOLLIDON^®^ 90 F > KOLLIDON^®^ 25) [45], and possibly the potential for H-bond interaction with HPMC’s -OH groups in the lactam parts of the PVP molecule. The PVPs differ from PEG/PEO in molecular weight, the viscosity of solution [43,44,45], and particle size (see Appendix A), which could all potentially influence the variability of carvedilol release. Drug release from HPMC matrices, which include additional polymer, can be influenced by weak interpolymer H-bonds and the ability of the additional polymer to form H-bonds with HPMC molecules in a dry state and in the presence of water. PVPs are less hydrophilic than PEG/PEO (and HPMC), and the hydrophilic amide moieties of a PVP molecule, with a potential for H-bond interaction with HPMC, are sterically hindered by the hydrophobic carbon chain [46], which are consequences of differences in the molecular structures among these excipients. The carvedilol release profile of the KOLLIDON^®^ 25 formulation is only partially similar to both PEG formulations due to the high variability of carvedilol release in the KOLLIDON^®^ 25 formulation (see Appendix A, Figure 3). The carvedilol release profile variability in the KOLLIDON^®^ 25 formulation is the highest among all tested formulations (see Figure 3, Figure 4, Figure 5, Figure 6 and Figure 7, Appendix A), and mannitol formulations come in a close second by also exhibiting high intra-batch carvedilol release profile variability.

##### Mannitol and Lactose

Mannitol

In the mannitol group, both spray-dried mannitol formulations, Parteck^®^ M 100 and Parteck^®^ M 200, demonstrate faster carvedilol release than crystalline mannitol formulations, C*Pharm Mannidex 16700 and PEARLITOL^®^ 160C (see Figure 4a, Appendix A). Release profiles of both spray-dried mannitol formulations are similar to each other, and both crystalline mannitol formulations are similar to each other according to the f_2_ metric. However, no significant similarity exists in carvedilol release profiles between any two release profiles where one comes from the spray-dried mannitol group and the other from the crystalline mannitol group of formulations according to the f_2_ metric (see Appendix A). On average, the carvedilol release profile of the Parteck^®^ M 100 formulation is somewhat faster than that of the Parteck^®^ M 200 one (see Figure 4a, Appendix A), which can be attributed to a smaller spray-dried particle size of Parteck^®^ M 100 in comparison to Parteck^®^ M 200 (see Appendix A). In contrast, the average carvedilol release profile of the smaller mannitol particle size C*Pharm Mannidex 16700 formulation is slower, although not in a practically significant way, than that of PEARLITOL^®^ 160C one (see Appendix A, Figure 4a). Differences in average carvedilol release kinetics obtained with different grades of mannitol, especially differences between crystalline and spray-dried grades of mannitol, could be attributed to differences in the sorbitol content (see Table 2) and to the larger specific surface area of particles of spray-dried mannitol grades [47]. In addition, there are differences among the mannitol formulations in incorporated mannitol grades’ particle size (see Appendix A), achieved tablet hardness, and mechanical behaviour of compression mixtures during compression, which influences tablet porosity. Both spray-dried mannitol formulations exhibit T_lag_, whereas crystalline PEARLITOL^®^ 160C exhibits a positive F_0_ (indication of some burst release) (see Figure 2). With the C*Pharm Mannidex 16700 formulation, it is hard to say whether a burst effect is observed since the variability of carvedilol release is very high at the beginning; however, data show there is no T_lag_. Observed T_lag_ in spray-dried mannitol formulations could be a result of high tablet hardness achieved during compression but also due to both spray-dried grades of mannitol producing faster carvedilol release profiles. This insinuates faster mannitol dissolution together with faster initial HPMC hydration and swelling, which quickly starts to control carvedilol release. Mannitol alone exhibits a high apparent intrinsic dissolution rate (IDR) [48]. This high dissolution rate of mannitol, together with its osmotic effect, could facilitate a fast initial ‘salting in’ effect on HPMC, promoting initial hydration and swelling of HPMC and, thus, the onset of controlled carvedilol release [4]. On the other hand, mannitol could also then ‘salt out’ HPMC by effectively ‘stealing’ available water from HPMC locally and hindering its hydration and swelling. This would hinder HPMC’s ability to adequately control carvedilol release, which could result in a relatively high carvedilol release rate and increased variability in carvedilol release among individual tablets as HPMC would not be able to form as efficient ‘gel’ layer as without a salting-out entity being present in the tablet [4]. Furthermore, mannitol, being a small molecule, does not produce a high viscosity of solution (such as PEG/PEO or PVP and also lower than that of lactose and sucrose), thus facilitating fast diffusion of the dissolved mannitol and carvedilol out of pores and capillaries during dissolution [43,44,45,49]. All the mentioned factors could explain why a relatively low burst effect indication or even T_lag_ is observed in the mannitol group in comparison to some grades of lactose (the lactose group will be discussed later), and at the same time, fast carvedilol release profiles and high carvedilol release profile variability (see Figure 2 and Figure 4, Appendix A).

The carvedilol release profile of the Parteck^®^ M 100 formulation is similar to both PEG formulations, whereas the Parteck^®^ M 200 formulation exhibits only partial similarity according to the f_2_ metric (see Appendix A). Both spray-dried mannitol formulations exhibit higher carvedilol release profile variability than PEG formulations (see Figure 3a and Figure 4a, Appendix A). The release of carvedilol is generally a little slower in spray-dried mannitol formulations in comparison to PEG formulations (see Appendix A). Spray-dried mannitol formulations are also similar to Polyglykol^®^ 8000 P and POLYOXᵀᴹ WSR N-80 formulations in T_lag_ occurrence (see Figure 2). The carvedilol release profile of the crystalline mannitol C*Pharm Mannidex 16700 formulation is similar to that of the POLYOXᵀᴹ WSR N-80 one, whereas the crystalline mannitol PEARLITOL^®^ 160C formulation exhibits only partial similarity with the PEO formulation according to the f_2_ metric (see Appendix A). The carvedilol release profile of the POLYOXᵀᴹ WSR N-80 formulation is somewhat slower in comparison to both crystalline mannitol formulations (see Appendix A). The PEO formulation, however, exhibits significantly lower carvedilol release profile variability among individual tablets (see Figure 3a and Figure 4a, Appendix A). It also clearly demonstrates T_lag_, whereas crystalline mannitol formulations do not. In contrast, the PEARLITOL^®^ 160C formulation demonstrates some burst release; however, in the case of the C*Pharm Mannidex 16700 one, this is less clearly evident due to higher carvedilol release variability at the beginning of the carvedilol release profile (see Figure 2).

All of the mannitol formulations exhibit partial similarity with the KOLLIDON^®^ 25 formulation (see Appendix A) according to the f_2_ metric; the similarity among carvedilol release profiles is only partial due to the high variability of carvedilol release observed in all of these formulations (see Figure 3b and Figure 4a, and Appendix A). The KOLLIDON^®^ 25 formulation demonstrates generally faster carvedilol release than crystalline mannitol formulations, whereas spray-dried mannitol formulations release carvedilol faster in the 2nd half of the release profile (see Appendix A). In addition, the KOLLIDON^®^ 25 formulation does not exhibit T_lag_, whereas spray-dried mannitol formulations do (see Figure 2).

2.Lactose

In carvedilol release profiles of the lactose group of formulations, two clusters can be identified. The first cluster consists of formulations containing Lactochem^®^ Crystals (a crystalline form of lactose), SuperTab^®^ 11SD (a spray-dried form of lactose), and Tablettose^®^ 70 (an agglomerated form of lactose). The second cluster consists of Lactochem^®^ Fine Powder (contains milled crystals of lactose and exhibits the smallest particle size of all the tested grades of lactose) and FlowLac^®^ 100 (a spray-dried form of lactose with a similar particle size as SuperTab^®^ 11SD). Lactose grades in the first cluster demonstrate faster carvedilol release profiles than grades in the second cluster and a significantly greater burst effect, although the burst effect in the second cluster is also clearly present (see Figure 2 and Figure 4b). The burst effect in the first cluster is the highest among all tested formulations in our experiments. In each cluster, carvedilol release profiles are similar to each other according to the f_2_ metric and different from the release profiles from the second cluster (see Appendix A). The variability of carvedilol release among individual tablets is much lower in both lactose clusters than that observed in the mannitol group, and the release rate of carvedilol is also much slower (see Figure 4, and Appendix A). Generally, lactose exhibits a lower apparent IDR than mannitol [48], so slower carvedilol release from tablets containing lactose in comparison to mannitol was expected. On the molecular level, lactose also contains more -OH groups per molecule than mannitol, which can potentially interact with HPMC -OH groups and stabilise carvedilol release more, thus producing less variability in carvedilol release among individual tablets than that observed in mannitol formulations. However, lactose does not have nearly as large H-bond interaction potential per single molecule as PEGs/PEOs, which could partially explain the difference in intra-batch carvedilol release variability between the two. Although lactose does not dissolve as fast as mannitol, it still dissolves relatively fast, has an osmotic effect, and facilitates the ingress of dissolution media into tablets. Its slower dissolution than that of mannitol [48] could result in less initial ‘salting in’ effect on HPMC and contribute to its high burst effect in comparison to other tested soluble excipients used in tested carvedilol formulations. It also exhibits a viscosity of the saturated solution, which is higher in comparison to mannitol but much lower than that of sucrose, PVPs, and PEGs/PEO [43,44,45,49]. This could help to explain the higher burst effect in lactose formulations at the beginning, but at the same time, better stabilization of carvedilol release in comparison to mannitol formulations after a saturated lactose solution is achieved. It seems that with lactose in general, unlike with other tested soluble excipients, any carvedilol particles present on the surface of tablets manage to dissolve before the HPMC can hydrate, swell, form a ‘gel’ layer, and start to control carvedilol release.

Comparing the performance of Lactochem^®^ Crystals and Lactochem^®^ Fine Powder, whose technological procedure of production is essentiality the same, apart from milling the produced lactose crystals in the case of Lactochem^®^ Fine Powder, a much slower carvedilol release from tablets containing Lactochem^®^ Fine Powder (see Figure 4b) can be observed. Smaller milled particles potentially result in smaller pores and capillaries while particles dissolve. Diffusion out of smaller diameter pores and capillaries is generally slower than out of larger ones. This is one factor that could explain why the Lactochem^®^ Fine Powder formulation exhibits slower carvedilol release than the Lactochem^®^ Crystals one. Faster carvedilol release with Lactochem^®^ Fine Powder than with Lactochem^®^ Crystals was expected, not just because of the difference in particle size (see Appendix A), but also because milled lactose usually contains amorphous lactose, at least on the surface of particles if not also in the majority of material obtained after milling [50,51,52,53,54].

Both spray-dried grades of lactose, SuperTab^®^ 11SD and FlowLac^®^ 100, exhibit surprisingly different carvedilol release profiles (see Appendix A, Figure 4b), although their particle size distribution is similar (see Appendix A).

Tablettose^®^ 70 produced a carvedilol release profile, which was among the fastest in the lactose group and very similar to that of the SuperTab^®^ 11SD formulation (see Figure 4b and Appendix A). Tablettose^®^ exhibits a relatively large specific surface area of agglomerated particles [55], which could contribute to fast carvedilol release from HPMC matrix tablets, among other factors.

While inspecting carvedilol release profiles obtained using different lactose grades, increased carvedilol release variability towards the end of the carvedilol release profile in the case of Lactochem^®^ Fine Powder and FlowLac^®^ 100 (see Figure 4b, and Appendix A) was noticed. This behaviour is generally unwanted as mechanical stress to tablets, which occurs in the GIT, is significant, and if (partial) failure of matrix tablet consistency during the dissolution test is observed, this means that these matrix tablets could potentially collapse in the in vivo environment of GIT.

The carvedilol release profile of the Lactochem^®^ Fine Powder formulation is partially similar to that of the KOLLIDON^®^ 90 F one according to the f_2_ metric (see Appendix A). The similarity of profiles is only partial, probably due to the larger inter-tablet carvedilol release variability observed in the KOLLIDON^®^ 90 F formulation (see Figure 3b and Figure 4b, and Appendix A).

##### Sucrose and Maltodextrin

GLUCIDEX^®^ 19 (a spray-dried form of maltodextrin) and Granulated sugar N°1 600 (sucrose with large crystalline particles) demonstrate the slowest carvedilol release out of all tested formulations using selected water-soluble excipients in tested carvedilol formulations (see Figure 3, Figure 4 and Figure 5 and Appendix A). Both carvedilol release profiles are similar according to the f_2_ metric and do not exhibit significant burst effect nor T_lag_ (see Appendix A and Figure 2). The carvedilol release profile variability among individual tablets is low and similar to that observed with lactose grades (see Figure 4b and Figure 5 and Appendix A). The GLUCIDEX^®^ 19 formulation did demonstrate some increase in carvedilol release variability after t = 480 min (8 h) in comparison to the Granulated sugar N°1 600 one. However, variability was still generally low and acceptable (see Figure 5 and Appendix A). This might have occurred due to the heterogenic composition of GLUCIDEX^®^ 19 [56]. The molecularly smaller components of maltodextrin, i.e., glucose and disaccharides, probably dissolve faster, whereas polysaccharides dissolve slower. This could potentially result in a change in maltodextrin composition during the dissolution of its components and could be a reason for the observed increased carvedilol release variability after tablets have been exposed to the dissolution media for some time. On the other hand, this heterogenic composition of maltodextrin, especially the presence of polysaccharides, which dissolve slower than mono and disaccharides and result in a relatively high viscosity of maltodextrin solution [57,58,59,60], is responsible for a relatively slow and stable carvedilol release profile considering maltodextrin is a soluble filler in an HPMC matrix tablet.

In contrast to GLUCIDEX^®^ 19, Granulated sugar N°1 600 is a well-defined homogenous substance containing crystalline sucrose with large crystals (see Appendix A). The large size of sucrose crystals in Granulated sugar N°1 600, together with the relatively high viscosity of a saturated solution [49], is responsible for its slow dissolution, producing a relatively slow and stable carvedilol release profile, considering sucrose is a soluble filler in an HPMC matrix tablet. These two factors seem to override the influence of a fast sucrose IDR, which is even faster than that of mannitol [48].

The carvedilol release profile of the Granulated sugar N°1 600 formulation is partially similar to that of the FlowLac^®^ 100 one (the FlowLac^®^ 100 formulation produced the slowest carvedilol release profile out of all tested lactose formulations), whereas the GLUCIDEX^®^ 19 formulation and the FlowLac^®^ 100 formulation exhibit similar carvedilol release profiles according to the f_2_ metric. In addition, the carvedilol release profile of the GLUCIDEX^®^ 19 formulation is partially similar to that of the Lactochem^®^ Fine Powder formulation (see Appendix A). A more detailed analysis reveals that the carvedilol release profiles obtained with FlowLac^®^ 100 and Lactochem^®^ Fine Powder formulations are still markedly faster than that obtained with sucrose and maltodextrin ones (see Appendix A). This observation is in line with previously mentioned literature data regarding the viscosity of saturated solutions of lactose, sucrose, and maltodextrin [49,57,58,59,60] and a significantly larger particle size of Granulated sugar N°1 600 than that of both mentioned lactose grades (see Appendix A). 

#### 3.1.3. Formulations Using Insoluble Excipients

##### Dibasic Calcium Phosphate and Pregelatinized Starch

The carvedilol release profiles of both DCP grades’ formulations are similar according to the f_2_ metric (see Appendix A). Looking in more detail, some differences between the two can be observed. The DI-CAFOS^®^ A12 formulation exhibits a lower burst effect than the EMCOMPRESS^®^ Anhydrous one (see Figure 2), so carvedilol release is faster at the beginning in the case of the EMCOMPRESS^®^ Anhydrous formulation. In contrast, towards the end of the carvedilol release profile, the DI-CAFOS^®^ A12 formulation demonstrates somewhat faster carvedilol release than the EMCOMPRESS^®^ Anhydrous one (see Figure 6 and Appendix A). DCP does not interact with HPMC or other constituents in the tablet in any significant way, so the observed carvedilol release variability can be considered as a ‘baseline’ variability, dependent more or less on the particle size of DCP. HPMC is more homogeneously distributed in tablets containing DI-CAFOS^®^ A12 simply because DI-CAFOS^®^ A12 contains smaller DCP particles than EMCOMPRESS^®^ Anhydrous (see Appendix A). This means that HPMC can swell more evenly at the beginning of carvedilol release leading to a lesser observed burst effect observed in the DI-CAFOS^®^ A12 formulation in comparison to the EMCOMPRESS^®^ Anhydrous one. However, the smaller DCP particles of DI-CAFOS^®^ A12 erode more readily than larger DCP particles of EMCOMPRESS^®^ Anhydrous, which explains why towards the end of the release profile, more carvedilol is released from the DI-CAFOS^®^ A12 formulation than from EMCOMPRESS^®^ Anhydrous one.

Carvedilol release profiles produced using STARCH 1500^®^ ↓PS and STARCH 1500^®^ ↑PS are similar according to the f_2_ metric (see Appendix A). A more detailed analysis reveals that the STARCH 1500^®^ ↓PS formulation exhibits a larger burst effect and more carvedilol release variability at the beginning of carvedilol release in comparison to the STARCH 1500^®^ ↑PS one (see Figure 2 and Figure 6 and Appendix A). This cannot be explained by the difference in particle size between the two STARCH 1500^®^ samples (see Appendix A) and might be due to differences between both samples in the composition of three types of particles formed during STARCH 1500^®^ hydration, as described by Wong [27].

The DI-CAFOS^®^ A12 formulation’s carvedilol release profile is partially similar to that of the STARCH 1500^®^ ↓PS formulation (see Appendix A). A more detailed analysis shows that the carvedilol release profile of the DI-CAFOS^®^ A12 formulation is generally faster than that of STARCH 1500^®^ ↓PS one and also exhibits less burst effect (see Figure 2 and Figure 6, Appendix A). The EMCOMPRESS^®^ Anhydrous formulation’s carvedilol release profile is similar to that of the STARCH 1500^®^ ↓PS’ one according to the f_2_ metric (see Appendix A), although it is somewhat faster and also exhibits less burst effect (see Figure 2 and Figure 6, Appendix A). The carvedilol release of the EMCOMPRESS^®^ Anhydrous formulation is also partially similar to that of the STARCH 1500^®^ ↑PS one (see Appendix A). However, the carvedilol release is somewhat faster in the case of the EMCOMPRESS^®^ Anhydrous formulation (see Figure 6 and Appendix A), while both formulations exhibit a similar burst effect (see Figure 2).

##### MCC and EC

The carvedilol release profiles of AVICEL^®^ PH-102, AVICEL^®^ PH-200, and ETHOCELᵀᴹ Standard 20 Premium formulations are similar according to the f_2_ metric (see Appendix A). They all demonstrate very low variability of carvedilol release among individual tablets throughout the entire carvedilol release profile and exhibit virtually no burst effect (see Figure 2 and Figure 7, Appendix A). The AVICEL^®^ PH-200 and the ETHOCELᵀᴹ Standard 20 Premium formulations’ carvedilol release rate is practically the same and is somewhat faster than that of AVICEL^®^ PH-102 (see Appendix A). MCC is known to swell, form channels, and facilitate water transport from the outer media into the HPMC matrix [61], where this water is then available for HPMC swelling not only on the surface of the tablet but also deeper within the tablets. This is probably the reason for the observed slow and controlled carvedilol release in both MCC formulations; MCC effectively facilitates controlled and sufficient water availability for HPMC, which can efficiently hydrate, swell and form a ‘gel’ layer, thus, producing a slow and stable carvedilol release. AVICEL^®^ PH-200 contains larger particles than AVICEL^®^ PH-102 (see Appendix A) and, thus, forms larger channels, which facilitates faster diffusion of carvedilol from the tablets into the surrounding media than in the case of AVICEL^®^ PH-102. This explains the somewhat faster carvedilol release profile of the AVICEL^®^ PH-200 formulation in comparison to the AVICEL^®^ PH-102 one. The before-mentioned channel formation and water transport facilitation via MCC are probably responsible for low carvedilol release variation among individual tablets as well as for virtually non-present burst effect (see Figure 2 and Figure 7, Appendix A). A comparable and not slower carvedilol release profile using EC instead of MCCs was surprising as EC can also be used as a hydrophobic matrix former and is a well-established hydrophobic coating agent. In our experimental system, the carvedilol release performance of EC and both MCCs was practically the same and observed small differences are of no practical significance (see Figure 2 and Figure 7, Appendix A).

The carvedilol release profile of the AVICEL^®^ PH-102 formulation is partially similar to that of the EMCOMPRESS^®^ Anhydrous one and similar to that of both STARCH 1500^®^ formulations (see Appendix A), although the carvedilol release profile of the EMCOMPRESS^®^ Anhydrous formulation is somewhat faster than that of the other three formulations (see Appendix A). Both the AVICEL^®^ PH-200 and the ETHOCELᵀᴹ Standard 20 Premium formulations are similar to both DCP formulations, both STARCH 1500^®^ formulations, and to the AVICEL^®^ PH-102 formulation regarding the carvedilol release profile (see Appendix A). A more detailed analysis shows that both STARCH 1500^®^ formulations and the AVICEL^®^ PH-102 formulation produce the slowest carvedilol release profiles, followed by the AVICEL^®^ PH-200 and the ETHOCELᵀᴹ Standard 20 Premium formulations, then by the EMCOMPRESS^®^ Anhydrous formulation and, finally, by the DI-CAFOS^®^ A12 formulation, which produces the fastest carvedilol release profile among all of the tested insoluble excipients’ formulations (see Appendix A). In all of the cases mentioned in this paragraph, the MCC and EC formulations are superior to the DCP and STARCH 1500^®^ formulations in that they vary less in carvedilol release (see Appendix A) and exhibit less, i.e., practically no, burst release (see Figure 2).

## 4. Conclusions

The choice of excipients used in HPMC matrix systems is of great importance for in vitro drug release performance. Information from the literature that indicates that soluble excipients generally produce faster carvedilol release profiles than insoluble ones was confirmed. However, the broad range of obtained carvedilol release profiles using different soluble excipients was surprising. Most importantly, significant differences in carvedilol release variation, burst release, and lag time not only among chemically different entities of tested excipients but also within different grades of excipients were shown. This has great implications for the quality of HPMC matrix tablets in general as it demonstrates that lowering the drug release variation, as well as all but excluding the occurrence of burst release and lag time, can be obtained by choosing the appropriate excipient to be used as a ‘filler’ in HPMC matrix tablets. Obtained results demonstrate that a simple change in the selected filer/drug release modifier can result in a large change in the carvedilol release profile. These results have broad scientific and industrial implications regarding using different fillers/drug release modifiers for optimising drug release from HPMC-based matrix solid dosage forms. Considering obtained results, PEGs and PEOs (in the lower molecular weight range for PEOs) can be identified as good soluble excipient choices for use in HPMC matrix tablets for obtaining a fairly fast carvedilol prolonged release profile. Specifically, Polyglykol^®^ 8000 P and POLYOXᵀᴹ WSR N-80 demonstrated no burst effect and very low variability in carvedilol release among individual tablets. Among insoluble excipients, MCC and EC seem to be the best choice. They demonstrate no burst effect and very low variability in carvedilol release among individual tablets. MCC is preferred, however, due to its superior tabletability and generally good processability in direct compression.

The similarity factor (f_2_) is an established and useful tool for the determination of the overall similarity of any two drug release profiles. In addition, the utilization and usefulness of LOESS as a general drug release profile modelling technique was demonstrated, which is a novel application of this technique. It facilitates detailed analysis of differences among the drug release profiles as well as the identification and estimation of burst release and lag time without the need to search for a suitable mathematical drug release model for each different formulation, i.e., drug release profile, which produces a sufficient fit with the experimental data to be practically useful in drug release profile analysis. LOESS modelling can be successfully applied to a wide variety of drug release profiles and utilized in the analysis of drug release if enough experimentally measured data points are available to facilitate good local regression fitting.

## Figures and Tables

**Figure 1 pharmaceutics-15-01525-f001:**
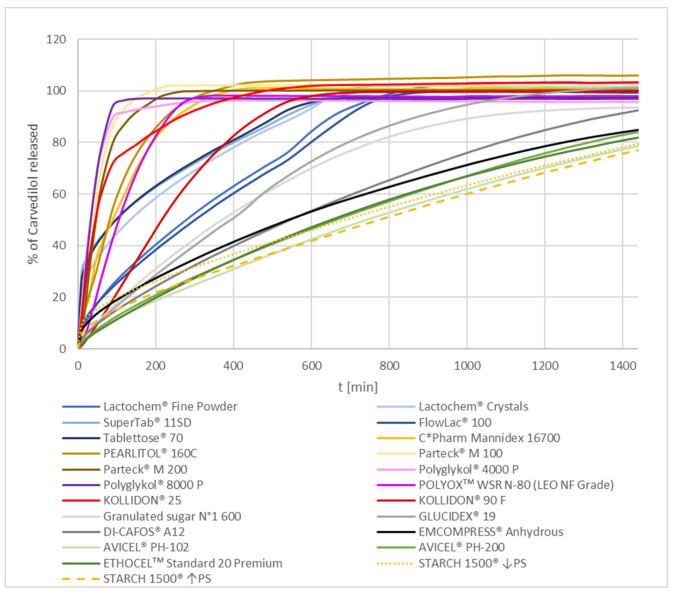
Mean carvedilol release from METHOCELᵀᴹ K15M Premium matrix tablets containing different selected excipients as carvedilol release modifiers.

**Figure 2 pharmaceutics-15-01525-f002:**
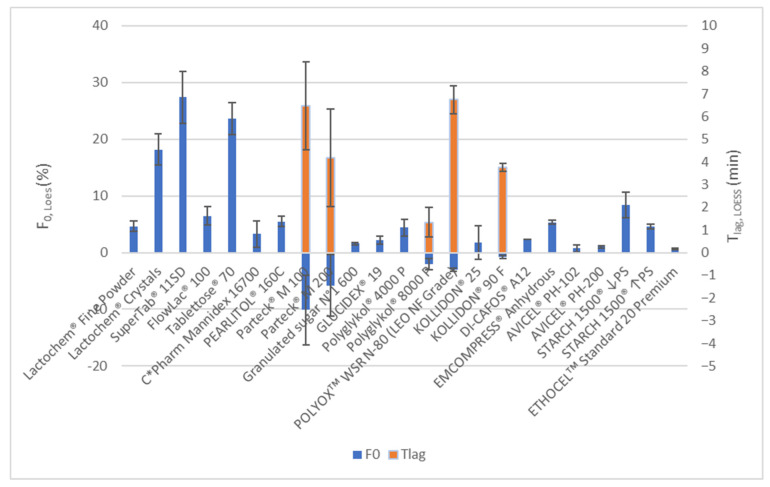
LOESS estimations of carvedilol burst release (F_0_ where F_0_ > 0%) and release onset lag time (T_lag_) for formulations using different selected excipients in METHOCELᵀᴹ K15M Premium-based matrix tablets. Mean values ± 1 SD (error bars) are presented in the graph. The T_lag_ results are shown only for experiments where a significant negative F_0_ was estimated.

**Figure 3 pharmaceutics-15-01525-f003:**
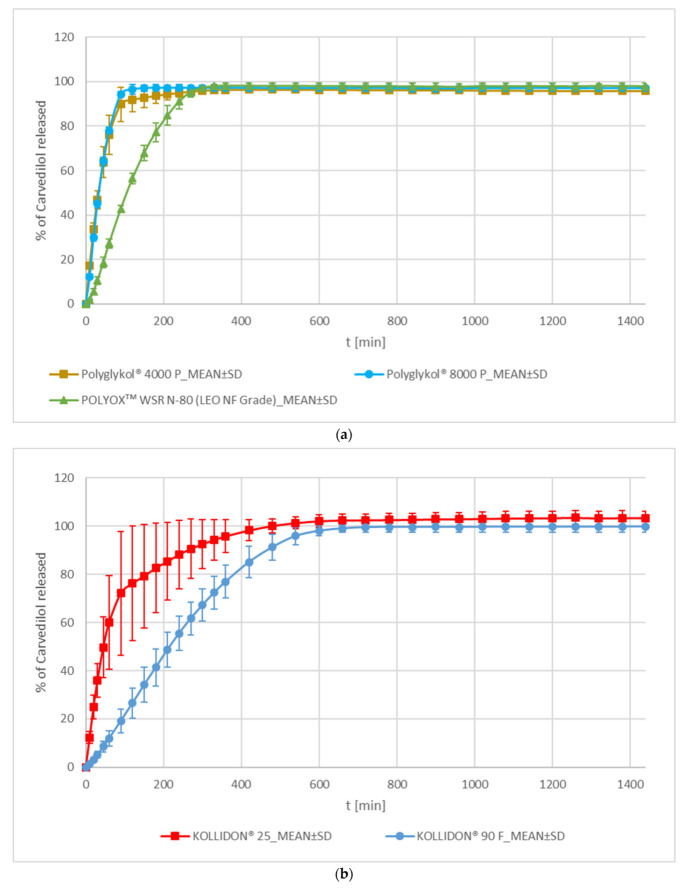
Carvedilol release from METHOCELᵀᴹ K15M Premium matrix tablets containing PEG/PEO (**a**) or PVP (**b**) as carvedilol release modifiers. Full lines with error bars represent mean carvedilol release ± 1 SD.

**Figure 4 pharmaceutics-15-01525-f004:**
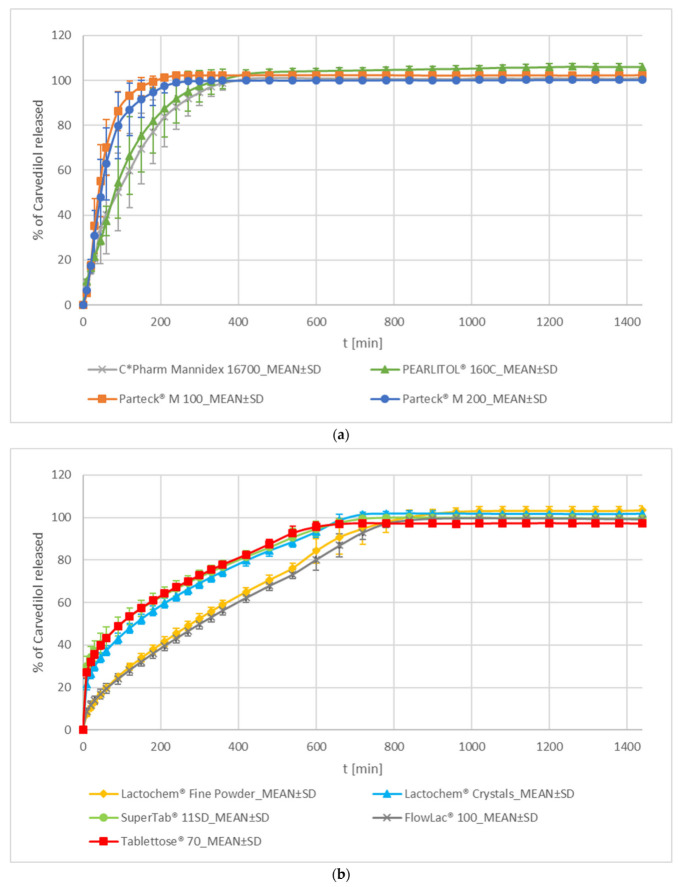
Carvedilol release from METHOCELᵀᴹ K15M Premium matrix tablets containing mannitol (**a**) or lactose (**b**) as carvedilol release modifiers. Full lines with error bars represent mean carvedilol release ± 1 SD.

**Figure 5 pharmaceutics-15-01525-f005:**
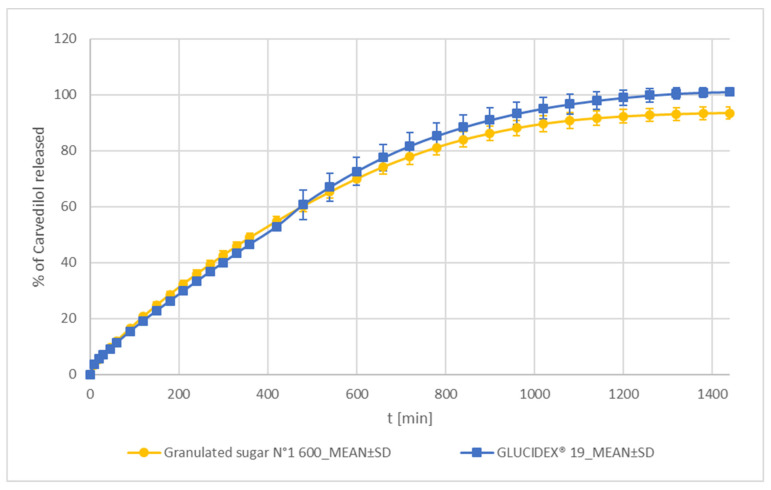
Carvedilol release from METHOCELᵀᴹ K15M Premium matrix tablets containing sucrose or maltodextrin as carvedilol release modifiers. Full lines with error bars represent mean carvedilol release ± 1 SD.

**Figure 6 pharmaceutics-15-01525-f006:**
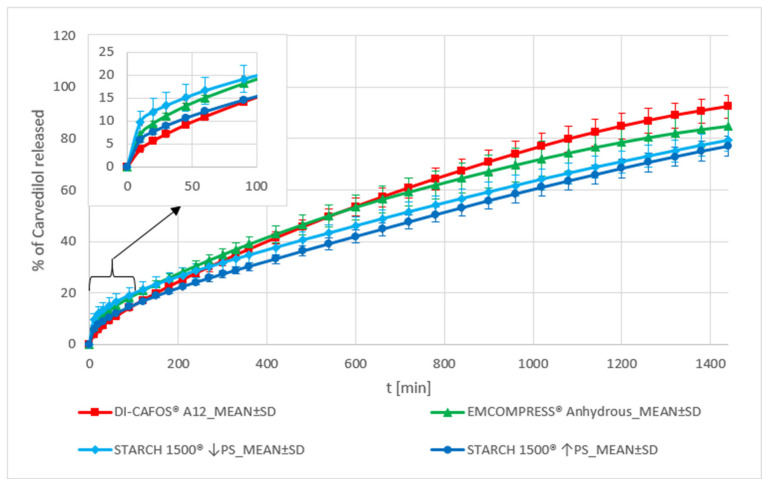
Carvedilol release from METHOCELᵀᴹ K15M Premium matrix tablets containing DCP or pregelatinized starch as carvedilol release modifiers. Full lines with error bars represent mean carvedilol release ± 1 SD.

**Figure 7 pharmaceutics-15-01525-f007:**
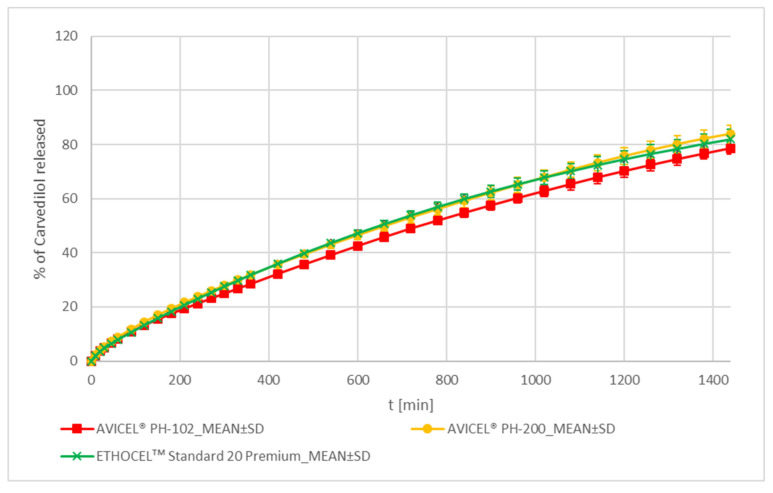
Carvedilol release from METHOCELᵀᴹ K15M Premium matrix tablets containing MCC or EC as carvedilol release modifiers. Full lines with error bars represent mean carvedilol release ± 1 SD.

**Table 1 pharmaceutics-15-01525-t001:** Composition of tablets.

Ingredient	Functionality of Ingredient	The Theoretical Amount of Ingredients in a Tablet (mg)	Theoretical *w*/*w* % of Ingredients in a Tablet	Additional Info
Carvedilol (free base) ^1^ (Ph. Eur.: Carvedilol)	Drug substance ^1^	64.8	10.00	see Appendix A
METHOCELᵀᴹ K15M Premium (HPMC 2208 with nominal viscosity 17,700 mPa·s; Ph. Eur.: Hypromellose)	Hydrophilic matrix forming agent	97.2	15.00	FRC data from CoA ^2^: viscosity 21,762 mPa·s ^3^, % of methoxy-groups (-OCH_3_, %MeO) 23.6% ^4^, % of hydroxypropoxygroups (-OCH_2_CH(OH)CH_3_, %HP) 9.6% ^4^; see Appendix A
See Table 2	Filler/Bulking agent/Carvedilol release modifier	475.632	73.40	see Appendix A
AEROSIL^®^ 200 Pharma (Colloidal silicon dioxide; Ph. Eur.: Silica, Colloidal Anhydrous)	Glidant	1.944	0.30	/
Magnesium stearate EUR PHAR Vegetable (Ph. Eur.: Magnesium stearate)	Lubricant	8.424	1.30	see Appendix A
Total		648.0	100.0	/

^1^ a model, poorly water-soluble drug, described in PhEur and USP as practically insoluble in water, ^2^ data regarding particle size were not provided in the CoA, ^3^ BROOKFIELD, 2% in water, T = 20 °C, ^4^ on a dry weight basis.

**Table 2 pharmaceutics-15-01525-t002:** Excipients used as water-soluble (a) or water-insoluble (b) fillers/bulking agents/carvedilol release modifiers in experiments. The solubility of water-insoluble excipients is negligible in comparison to water-soluble ones.

(a)				
**Excipient Generic Name**	**Selected Excipient’s Marketed Product Name**	**Name Abbreviation**	**Manufacturer**	**Additional Info ^1^**
Polyethylene Glycol (Ph. Eur.: Macrogols) & Polyethylene Oxide (Ph. Eur.: Macrogols, High-Molecular-Mass)	Polyglykol^®^ 4000 P	PEG 4k	Clariant Produkte (Deutschland) GmbH, Frankfurt, Germany	M = 4017 g/mol (CoA)
	Polyglykol^®^ 8000 P	PEG 8k		M = 8026 g/mol (CoA)
	POLYOXᵀᴹ WSR N-80 (LEO NF Grade)	PEO	DUPONT, Nutrition and Biosciences (Freienbach, Switzerland) GmbH	nominal M of 200,000 g/mol;
Povidone	KOLLIDON^®^ 25	PVP K25	BASF SE, Ludwigshafen, Germany	K value = 24.7 (CoA)
	KOLLIDON^®^ 90 F	PVP K90		K value = 92.4 (CoA)
Mannitol	C*Pharm Mannidex 16700	MAN_C_1	Cargill S.r.l., Milan, Italy	crystalline D-mannitol, sorbitol content = 0.6% (CoA)
	PEARLITOL^®^ 160C	MAN_C_2	ROQUETTE Frères, Lestrem, France	crystalline D-mannitol, sorbitol content = 0.7% (CoA)
	Parteck^®^ M 100	MAN_SD_1	Merck KGaA, Darmstadt, Germany	spray-dried D-mannitol, sorbitol content = 1.3% (CoA)
	Parteck^®^ M 200	MAN_SD_2		spray-dried D-mannitol, sorbitol content = 1.3% (CoA)
Lactose Monohydrate	Lactochem^®^ Crystals	LAC_M	DFE Pharma GmbH and Co. KG, Goch, Germany	crystalline lactose monohydrate
	Lactochem^®^ Fine Powder	LAC_C		milled lactose monohydrate
	SuperTab^®^ 11SD	LAC_SD_1		spray-dried lactose monohydrate
	FlowLac^®^ 100	LAC_SD_2	MEGGLE GmbH and Co. KG, Wasserburg, Germany	spray-dried lactose monohydrate
	Tablettose^®^ 70	LAC_AG		agglomerated lactose monohydrate
Sucrose	Granulated sugar N°1 600	SUC	Tereos, Moussy-le-Vieux, France	crystalline sucrose
Maltodextrin	GLUCIDEX^®^ 19	MD_SD	ROQUETTE Frères, France	a spray-dried mixture of glucose, disaccharides, and polysaccharides
(b)				
**Excipient Generic Name**	**Selected Excipient’s Marketed Product Name**	**Name Abbreviation**	**Manufacturer**	**Additional Info ^1^**
Anhydrous Dibasic Calcium Phosphate (Ph. Eur.: Calcium Hydrogen Phosphate)	DI-CAFOS^®^ A12	DCP_1	Chemische Fabrik Budenheim KG, Budenheim, Germany	anhydrous DCP
	EMCOMPRESS^®^ Anhydrous	DCP_2	JRS Pharma GmbH and Co. KG, Rosenberg, Germany	anhydrous DCP
Microcrystalline Cellulose (Ph. Eur.: Cellulose, Microcrystalline)	AVICEL^®^ PH-102	MCC 101	DuPont Nutrition Ireland, Little Island, Ireland	nominal particle size of app. 100 µm
	AVICEL^®^ PH-200	MCC 200		nominal particle size of app. 180 µm
Ethylcellulose	ETHOCELᵀᴹ Standard 20 Premium	EC	DOW, Specialty Electronic Materials Switzerland GmbH, Horgen, Switzerland	FRC data from CoA: viscosity 20.6 mPa·s, ethoxyl content (assay) 48.7%
Pregelatinized Starch	STARCH 1500^®^ sample with smaller particle size (↓PS)	PS_1	Colorcon Inc., Harleysville, PA, USA	sieve analysis: 99.3% through 100 mesh and 46.9% through 270 mesh (CoA)
	STARCH 1500^®^ sample with larger particle size (↑PS)	PS_2		sieve analysis: 93.5% through 100 mesh and 31.9% through 270 mesh (CoA)

^1^ for particle size analysis information, refer to Appendix A.

## Data Availability

Not applicable.

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
