# Peer review of "Comparative Study of Selected Excipients’ Influence on Carvedilol Release from Hypromellose Matrix Tablets"

_pharmaceutics, 2023, doi:10.3390/pharmaceutics15051525_

Round 1
Reviewer 1 Report (Previous Reviewer 2)
The manuscript titled “Comparative study of selected excipients’ influence on carvedilol release from hypromellose matrix tablets” has been reviewed. This manuscript is well written and all the results are well presented. I have few queries as;
· In material section, purity of carvidolol powder with respect to pharmacopoeial standard needs to be mentioned
· Why carvidalol was selected as model drug? Selection of model drug needs to be justified in introduction section.
· Compression of the tablet is very high as compared to its strength and polymer quantity which is ambiguous. Such huge quantities of diluent will definitely affect release characteristics of the drug. It needs to be properly addressed
· Tablets were compressed using 12 mm round punches. Proper design of punches (flat, concave or with beveled edges etc) needs to be mentioned as shape of tablet has an impact on its characteristics.
· Mechanical strength of tablets have significant effect on release characteristics. Furthermore mechanical strength depends upon characteristics of diluent and other excipeints. Although tablets were compressed under constant compression force but their crushing strength needs to be determined and correlated with release characteristics.
I recommend to accept the manuscript for publication, after incorporation of the suggested changes and properly addressing the queries.
The manuscript titled “Comparative study of selected excipients’ influence on carvedilol release from hypromellose matrix tablets” has been reviewed. This manuscript is well written and all the results are well presented. I have few queries as;
· In material section, purity of carvidolol powder with respect to pharmacopoeial standard needs to be mentioned
· Why carvidalol was selected as model drug? Selection of model drug needs to be justified in introduction section.
· Compression of the tablet is very high as compared to its strength and polymer quantity which is ambiguous. Such huge quantities of diluent will definitely affect release characteristics of the drug. It needs to be properly addressed
· Tablets were compressed using 12 mm round punches. Proper design of punches (flat, concave or with beveled edges etc) needs to be mentioned as shape of tablet has an impact on its characteristics.
· Mechanical strength of tablets have significant effect on release characteristics. Furthermore mechanical strength depends upon characteristics of diluent and other excipeints. Although tablets were compressed under constant compression force but their crushing strength needs to be determined and correlated with release characteristics.
I recommend to accept the manuscript for publication, after incorporation of the suggested changes and properly addressing the queries.
Author Response
Authors’ replies to reviewer’s comments:
»The manuscript titled “Comparative study of selected excipients’ influence on carvedilol release from hypromellose matrix tablets” has been reviewed. This manuscript is well written and all the results are well presented.”
Author response: We sincerely appreciate the time and effort that you dedicated to providing feedback on our manuscript and are grateful for the insightful comments on and valuable improvements to our paper.
“In material section, purity of carvedilol powder with respect to pharmacopoeial standard needs to be mentioned”
Author response: Thank you for pointing this out. All of the used materials complied with their Ph. Eur. specifications. We added this statement in revised manuscript at the end of the ‘2.1. Materials’ section.
“Why carvedilol was selected as model drug? Selection of model drug needs to be justified in introduction section.”
Author response: Thank you for your question. There is already an SR formulation of carvedilol intended for once-daily dosing, namely Coreg CR® extended-release capsules 80 mg, available in the market. This means that carvedilol is a suitable candidate for SR formulations. Since the marketed formulation is not an HPMC-based matrix system and comes in capsule dosage form, we decided to experiment with incorporating carvedilol into directly compressed HPMC-based matrix tablets.
“Compression of the tablet is very high as compared to its strength and polymer quantity which is ambiguous. Such huge quantities of diluent will definitely affect release characteristics of the drug. It needs to be properly addressed.”
Author response: Thank you for your comment. The main compression force of 20 kN was chosen based on preliminary experiments to achieve satisfactory friability of tablets. At this compression force, we achieved a satisfactory friability also with the STARCH 1500® formulation which demonstrated too high friability at lower compression forces. The same main compression force was used in all experiments as already published research indicated that the compression force is an important factor influencing drug release from HPMC matrix tablets. Consequently, the hardness of tablets differed from formulation to formulation. The tablet hardness was measured and evaluated but we did not find it relevant enough to be included in the revised version of the manuscript as it did not contribute enough to the interpretation of differences among the obtained carvedilol release profiles. The reported diluent quantity in the tablets was selected to observe the effect of different types of ‘diluents’ on the carvedilol release profile. At the same time, we wanted to use such a combination of diluent quantity and target tablet weight for the tablet that the tablets would still be of a reasonable size.
“Tablets were compressed using 12 mm round punches. Proper design of punches (flat, concave or with beveled edges etc) needs to be mentioned as shape of tablet has an impact on its characteristics.”
Author response: Thank you for pointing this out. We did not present the details of the tablet shape, other than the general tablet shape, which was round, and the diameter of tablets (12 mm), because we initially did not see this as relevant. The goal of the research was to investigate the influence of a comprehensive selection of excipients on drug release from an HPMC matrix tablet within the same experimental setup and compare the obtained drug release profiles in a detailed way. The used tablet punches were the same for all produced tablets which was key for our goal of maintaining the same experimental setup throughout the research. However, based on your comment, we added more detailed information regarding the tablet shape in section ‘2.2.2. Production of tablets’ in the revised version of the manuscript.
“Mechanical strength of tablets have significant effect on release characteristics. Furthermore mechanical strength depends upon characteristics of diluent and other excipients. Although tablets were compressed under constant compression force but their crushing strength needs to be determined and correlated with release characteristics.”
Author response: The same main compression force was used in all experiments as already published research indicated that the compression force is an important factor influencing drug release from HPMC matrix tablets. Consequently, the hardness of tablets differed from formulation to formulation. The tablet hardness was measured and evaluated but we did not find it relevant enough to be included in the revised version of the manuscript as it did not contribute enough to the interpretation of differences among the obtained carvedilol release profiles. In the revised version of the manuscript we only mentioned the possible influence of the relatively high hardness of tablets produced using spray-dried mannitol grades Parteck® M 100 and Parteck® M 200 on carvedilol release lag-time occurrence.

Reviewer 2 Report (Previous Reviewer 3)
All the comments are well addressed. The revised version of the manuscript can be accepted for publication.
Author Response
Authors’ replies to reviewer’s comments:
We sincerely appreciate the time and effort that you dedicated to providing feedback on our manuscript.

Reviewer 3 Report (New Reviewer)
1. Is LOESS an abbreviation of some word in the abstract?
2. State the goal more clearly at the end of the introduction.
3. For all used excipients, and chemicals, specify in the section materials Pharmacopoeia name, or at least chemically.
4. Table 1 and 2 please try to improve visibility
5. Please use pharmacopoeial names
6. Chapter 2.2.2. you do not state the mass of carvedilol per tablet, considering its low therapeutic dose, you have an extremely high mass of tablets, this is honestly a drawback, which can reduce compliance. You have 30 times the weight of the tablet, compared to the dose of the active substance.
7. What is the pH-dependent solubility of carvedilol? In biorelevant conditions, it will be the first to be found in the stomach that we simulate with diluted HCl and pH 1.2. Would that be where the tablets would dissolve? It appears to me to have pH-dependent solubility, check and I'm afraid you've picked the pH where the solubility is the least, thus giving you the extended-release results you got. Check, think, always look at what the pharmacopeia says, always think when a substance is "complex" with pH-dependent solubility as you choose a medium. Use biorelevant media and devices. The tablet will spend 30-120 min in an acidic medium of pH 1.2, maybe everything will dissolve there, think about it.
8. A large part of chapter 3.1.1. I don't think it's in the right place
9. Add standard deviations to the dissolution profiles
10. Avoid writing in the first person plural
11. That's why you write the names of some excipients in caps lock
12. Figure 3 c you have in some points a huge deviation, ie sd, eg for aKollidon of 45-100% in one point. You will agree that this is not a very small difference... And it is high sd all the time, check these results, please. And in Figure 4a, so the calculation of the similarity factor is questionable. Because a single point that has a huge range of different results is not for comparison with other results
.
Author Response
Authors’ replies to reviewer’s comments:
»Is LOESS an abbreviation of some word in the abstract?”
Author response: Thank you for your question. Yes, ‘LOESS’ is an abbreviation for ‘Local (Weighted) Regression’. We explained this abbreviation in the introduction section of the manuscript. However, we found it a bit lengthy to explain the ‘LOESS’ abbreviation in the ‘Abstract’. We are aware that not everybody is familiar with this abbreviation. However, we feel that if the content of the ‘Abstract’ is intriguing enough for the individual reader, he or she will read through the article where the explanation of the ‘LOESS’ abbreviation can be found in the introduction section, where it is first used in the body of the article.
“State the goal more clearly at the end of the introduction.”
Author response: Thank you for pointing this out. We added this information in the last paragraph of the introduction section of the revised manuscript.
“For all used excipients, and chemicals, specify in the section materials Pharmacopoeia name, or at least chemically.”
Author response: Thank you for pointing this out. We added the information of materials’ Ph. Eur. names as a reference in Tables 1 and 2 of the revised manuscript.
“Table 1 and 2 please try to improve visibility.”
Author response: Thank you for your comment. We will check this in the final version of the manuscript and consult with the Pharmaceutics’ editorial team.
“Please use pharmacopoeial names.”
Author response: Thank you for your comment. We added the information of materials’ Ph. Eur. names as a reference in Tables 1 and 2 of the revised manuscript. However, throughout the manuscript, we mostly use selected excipients’ marketed product names, as there are differences in different grades of selected excipients regarding their influence on carvedilol release. In Tables 1 and 2 all of the names of materials mentioned throughout the manuscript are summarised for reference to the reader.
“Chapter 2.2.2. you do not state the mass of carvedilol per tablet, considering its low therapeutic dose, you have an extremely high mass of tablets, this is honestly a drawback, which can reduce compliance. You have 30 times the weight of the tablet, compared to the dose of the active substance.”
Author response: Thank you for your comment. The mass of carvedilol per tablet is stated in Table 1 in section ‘2.1. Materials’ prior to section ‘2.2.2. Production of tablets’. We did not state the mass of carvedilol per tablet in section ‘2.2.2. Production of tablets’ to avoid redundancy / repeating of the already stated information.
“What is the pH-dependent solubility of carvedilol? In biorelevant conditions, it will be the first tobe found in the stomach that we simulate with diluted HCl and pH 1.2. Would that be where thetablets would dissolve? It appears to me to have pH-dependent solubility, check and I'm afraidyou've picked the pH where the solubility is the least, thus giving you the extended-releaseresults you got. Check, think, always look at what the pharmacopeia says, always think when a substance is "complex" with pH-dependent solubility as you choose a medium. Use biorelevant media and devices. The tablet will spend 30-120 min in an acidic medium of pH 1.2, maybe everything will dissolve there, think about it.”
Author response: Thank you for pointing this out. The carvedilol solubility is indeed pH dependent but that is not all. According to Košir et al., 2018 (see citation below) the acetate buffer solution (pH = 4.5) was found to be the most discriminatory medium for studying carvedilol release from HPMC matrix tablets. According to in-house data, the solubility of carvedilol in acetate buffer solution (pH = 4.5) is the highest of all tested buffer solutions in the pH range from 1 to 8 (HCl media, acetate buffer media, phosphate buffer media, purified water media). Using acetate buffer with pH = 4.5 ensures sink conditions during dissolution testing.
Košir, Darjan, et al. "A study of critical functionality-related characteristics of HPMC for sustained-release tablets." Pharmaceutical Development and Technology 23.9 (2018): 865-873.
“A large part of chapter 3.1.1. I don't think it's in the right place”
Author response: Thank you for your comment. After careful consideration, we agreed with your concern and moved the first two paragraphs of chapter 3.1.1 to the introduction section of the revised manuscript.
“Add standard deviations to the dissolution profiles”
Author response: Thank you for your comment. The standard deviations are already included in Figures 3, 4, 5, 6 and 7, which provide more detailed information about the obtained carvedilol release profiles. In Figure 1, we intentionally left out the standard deviations and charted only the mean carvedilol release results, because adding the standard deviations to Figure 1 would make the chart too crowded. Figure 1 is part of the overview of the results presented in Chapter 3.1.1. and is therefore not intended to provide detailed information about the obtained carvedilol release profiles. As mentioned, this information is provided in Figures 3-7.
“Avoid writing in the first person plural”
Author response: Thank you for pointing this out. We will do our best to consider this comment in the revised version of the manuscript.
“That's why you write the names of some excipients in caps lock”
Author response: Thank you for your comment. However, we are not sure what exactly you meant by your comment. Can you please elaborate?
“Figure 3 c you have in some points a huge deviation, ie sd, eg for a Kollidon of 45-100% in one point. You will agree that this is not a very small difference... And it is high sd all the time, check these results, please. And in Figure 4a, so the calculation of the similarity factor is questionable. Because a single point that has a huge range of different results is not for comparison with other results.”
Author response: Thank you for your comment. The Kollidon®25 (Povidone K25) formulation especially did indeed demonstrate a very high carvedilol release variability. The results are correct. There is no error in the results. As you pointed out, the variability is also high with mannitols. We agree with you, that the observed f2 is very un-informative in such cases as it considers only mean drug release but not also the variability of the drug release. However, the bootstrapped f2 considers the variability in drug release along with the mean drug release and consequently proves to be a much more useful tool for overall comparison of two drug release profiles, especially in such cases where one or both of the formulations exhibit high drug release variability.

Round 2
Reviewer 3 Report (New Reviewer)
.
This manuscript is a resubmission of an earlier submission. The following is a list of the peer review reports and author responses from that submission.
Round 1
Reviewer 1 Report
Abstract: In the present research, we tested the influence of
selected excipients on carvedilol release performance from hydrophilic matrix tablets based on HPMC. Compression mixtures were directly compressed using constant compression speed and main compression force. We harnessed LOESS modelling to objectively and accurately estimate burst release, lag-time and compared times at which desired % of carvedilol was released from the tablets. The similarity between obtained carvedilol release profiles was estimated using the bootstrapped similarity factor (f₂). In the group of water-soluble carvedilol release modifying excipients, which produced relatively fast carvedilol release profiles, POLYOXᵀᴹ WSR N-80 and
Polyglykol® 8000 P demonstrated the best carvedilol release control and in
the group of water-insoluble carvedilol release modifying excipients, which
produced relatively slow carvedilol release profiles, AVICEL® PH-102 and
AVICEL® PH-200 performed best.
General comments
The paper deals with the formulation development at a scale that is not clearly declared. A table containing the exact composition of the formulation is missing.
The research work does not present enough novelty as for both the theoretical background and the experimental findings.
The introduction is too proliss and results in a confused picture, It should be simplified also because the majority of concept are already known.
The use of LOESS for fitting dissolution data is claimed novel. Instead the use of bootstrappped f2is already used in a regulatory setting. At any rate the estimated parameters of lag time and burst release have a limited transferability to an in vivo setting.
The discussion section should be divided in sub-sections. The core of the research is represented by the dissolution profiles of the various formulations. These are crowded in a single fig. which creates confusion. Please separate the various groups and provide separate comments.
The plots of the variability is redundant. Variability data should better be tabulated.
The results obtained that is the classification of the excipients in three classes is largely confirmatory of many published findings.
The scope of optimizing the fomulation is appreciated but the paper should be rewritten and reorganized avoiding redundancy and repetitions.
Reviewer 2 Report
Comments to author:
The manuscript titled “Comparative study of selected excipients’ influence on carvedilol release from hypromellose matrix tablets” has been reviewed. It is a well-designed study and different aspects of formulation development by direct compression have been properly discussed. I recommend to accept the manuscript for publication after some corrections as follows;
· Page # 4, Line # 168: LOESS should be defined at first appearance
· Page # 7, Homogenization of compression mixtures was performed at 32 rpm for 8 minutes before addition of lubricant and for 2 minute after addition of lubricant, using bi-conical mixer. How the homogenization tie was optimized. It needs to be mentioned. Fill volume of the blender needs to be mentioned.
· Page # 8, Line # 281: the sentence “The results of the characterization of tablets are presented in Figures 1, 2 and S10.” should be deleted.
· In all the figures (Figure 1 to 9) title of the figure is mentioned as part of the figure. It should be removed and details should be mentioned figure ligand
· It will be better to present the dissolution data, mentioned as Figure S11 in tabular form for better understanding and comparison
· Conclusion of the study is too lengthy and needs to be revised.
Reviewer 3 Report
The current manuscript focusing on investigating the effect of various water soluble and insoluble excipients on the release profiles of carvedilol matrix tablets is exciting and matches the journal's scope. The entire work is well-designed and executed. The authors are suggested to cross-check the entire manuscript for minor grammatical errors. The authors are suggested to address the below comments:
- The term "we" seems to be inappropriate in the abstract. Please modify as required. Also, check for the same throughout the manuscript.
- Please provide the information for the calibration curve.
- Why was dissolution performed for n=4?
- For figures 1 and 2 (legend), please correct the typo "hardness ± 1SD" and "average thickness ± 1SD."